# 18F-Fluciclovine Positron Emission Tomography in Prostate Cancer: A Systematic Review and Diagnostic Meta-Analysis

**DOI:** 10.3390/diagnostics11020304

**Published:** 2021-02-13

**Authors:** Giuseppina Biscontini, Cinzia Romagnolo, Chiara Cottignoli, Andrea Palucci, Fabio Massimo Fringuelli, Carmelo Caldarella, Francesco Ceci, Luca Burroni

**Affiliations:** 1Department of Nuclear Medicine, “Ospedali Riuniti” Hospital, 60126 Ancona, Italy; giuseppina.biscontini@ospedaliriuniti.marche.it (G.B.); chiara.cottignoli@ospedaliriuniti.marche.it (C.C.); andrea.palucci@spedaliriuniti.marche.it (A.P.); fabiomassimo.fringuelli@ospedaliriuniti.marche.it (F.M.F.); luca.burroni@ospedaliriuniti.marche.it (L.B.); 2Department of Nuclear Medicine, Fondazione Policlinico “A.Gemelli” IRCCS, 00168 Rome, Italy; carmelo.caldarella@policlinicogemelli.it; 3Department of Medical Sciences, Nuclear Medicine, University of Turin, 10126 Turin, Italy; francesco.ceci@unito.it

**Keywords:** 18F-Fluciclovine PET/CT, Prostate Cancer, lymph node staging, detection of relapse

## Abstract

Background: to explore the diagnostic accuracy of 18F-Fluciclovine positron-emission tomography (PET) in prostate cancer (PCa), considering both primary staging prior to radical therapy, biochemical recurrence, and advanced setting. Methods: A systematic web search through Embase and Medline was performed according to the Preferred Reporting Items for Systematic Reviews and Meta-analyses (PRISMA) guidelines. Studies performed from 2011 to 2020 were evaluated. The terms used were “PET” or “positron emission tomography” or “positron emission tomography/computed tomography” or “PET/CT” or “positron emission tomography-computed tomography” or “PET-CT” and “Fluciclovine” or “FACBC” and “prostatic neoplasms” or “prostate cancer” or “prostate carcinoma”. Only studies reporting about true positive (TP), true negative (TN), false positive (FP) and false negative (FN) findings of 18F-fluciclovine PET were considered eligible. Results: Fifteen out of 283 studies, and 697 patients, were included in the final analysis. The pooled sensitivity for 18F-Fluciclovine PET/CT for diagnosis of primary PCa was 0.83 (95% CI: 0.80–0.86), the specificity of 0.77 (95% CI: 0.74–0.80). The pooled sensitivity for preoperative LN staging was 0.57 (95% CI: 0.39–0.73) and specificity of 0.99 (95% CI: 0.94–1.00). The pooled sensitivity for the overall detection of recurrence in relapsed patients was 0.68 (95% CI: 0.63–0.73), and specificity of 0.68 (95% CI: 0.60–0.75). Conclusion: This meta-analysis showed promising results in term of sensitivity and specificity for 18F-Fluciclovine PET/CT to stage the primary lesion and in the assessment of nodal metastases, and for the detection of PCa locations in the recurrent setting. However, the limited number of studies and the broad heterogeneity in the selected cohorts and in different investigation protocols are limitation affecting the strength of these results.

## 1. Introduction

Prostate cancer (PCa) is the second most common cancer in the male population and represents the 15% of all cancers diagnosed [1,2] with a prevalence in elderly age of 59% (48–71%) [3]. Monitoring of the disease after radical prostatectomy (RP) is handled by measuring the Prostate Specific Antigen (PSA) serum levels, as defined by the definition of biochemical recurrence (BCR): PSA values above 0.2 ng/mL in patients treated with RP or PSA greater than 2 ng/mL above the nadir in patients treated with primary radiotherapy. However, the identification of the exact site(s) of the disease (pelvic vs. extra-pelvic recurrence) is crucial, and while PSA evaluation can optimally identify recurrent patients, it cannot provide anatomical functional information. The proper detection of the exact PCa location might significantly influence the therapeutic plan [4]. Recently, the use of positron emission tomography/computed tomography with several radiopharmaceuticals is gaining importance in this setting. In this scenario, 18F-Fluciclovine (18F-FACBC/1-amino-3fluorine 18F-fluorocyclobutane-1-carboxylic acid) has been proposed and has already received the Food and Drug Administration (FDA) and European Medicines Agency (EMA) approval for its use for PET imaging in biochemical recurrence of PCa. 18F-Fluciclovine is a synthetic amino acid transported across mammalian cell membranes by amino acid transporters such as LAT-1 and ASCT2. The upregulation of LAT-1 and ASCT2 activities in PCa is responsible for 18F-Fluciclovine uptake in PCa lesions [5]. After its uptake in the tumor cell, 18F-Fluciclovine is not furtherly metabolized. Thus, the intensity of uptake defined as “pathological” in PET image is highly dependent by the site and size of the lesion [6,7]. The 18F-Fluciclovine uptake in PCa cell lines is higher than that of methionine, glutamine, choline, and acetate [8]. Currently, 18F-flucicovine PET is recommended by the European Association of Urology (EAU) guidelines, in the case of BCR when prostate specific membrane antigen (PSMA) PET is not available, and the PSA serum level is higher than 1 ng/mL. While PSMA PET demonstrated superior diagnostic performance compared to choline PET and 18F-fluciclovine PET [9,10], this diagnostic procedure still holds limited availability in daily clinical practice. Furthermore, a non-negligible percentage of PCa phenotypes do not express PSMA (overall attested up to 5%). Accordingly, 18F-fluciclovine still represents a valid alternative to investigate recurrent PCa patients. Finally, present EAU guidelines do not suggest the use of new generation imaging to stage the disease prior to radical surgery. However, new emerging data strongly support its use in this scenario, namely in high-risk patients [11]. Therefore, the aim of this systematic review is to explore the diagnostic accuracy of 18F-Fluciclovine PET in PCa, both considering primary staging prior to radical therapy, biochemical recurrence, and advanced setting. A dedicated meta-analysis has been performed considering the meta-data derived by the reviewing process.

## 2. Materials and Methods

A systematic web search through Embase and Medline was performed according to the Preferred Reporting Items for Systematic Reviews and Meta-analyses (PRISMA) guidelines. Studies performed from 2011 to 2020 were evaluated. The terms used were “PET” or “positron emission tomography” or “positron emission tomography/computed tomography” or “PET/CT” or “positron emission tomography-computed tomography” or “PET-CT” and “Fluciclovine” or “FACBC” and “prostatic neoplasms” or “prostate cancer” or “prostate carcinoma”. Full-text publications in English were considered. The study population comprised male patients with histologically proven PCa. Studies using 18F-Fluciclovine in pre-surgery setting as well as biochemical recurrence and advanced setting were considered. Two investigators (C.C. and C.R.) independently performed the literature search. The inclusion criteria considered for each study included an assessment of true positive (TP), true negative (TN), false positive (FP), and false negative (FN) results of 18F-Fluciclovine PET or PET / CT for the diagnosis of primary cancer, lymph node staging and the detection of recurrence. Only studies providing these information were finally included in the study. The overall quality of the studies included was independently assessed by two authors, based on the 15 modified items Quality Assessment of Diagnostic Accuracy Studies (QUADAS-2, Table 1) [12]. Discrepancies between researchers were resolved with consensus.

## 3. Results

### 3.1. Patient Population

The whole abstract revision process is displayed in Figure 1. A total of 283 studies were screened. Forty-six were excluded after reviewing their content. A further 217 studies were subsequently excluded as the content was not considered relevant considering the aim of the analysis. Five studies were subsequently excluded due to insufficient data to evaluate sensitivity and specificity. In conclusion, 15 studies matched the inclusion criteria of the systematic review and were considered eligible for the meta-analysis. The number of subjects included in the final analysis were 697 in total. A multicentric study evaluated the diagnostic accuracy of 18F-Fluciclovine PET/CT in the diagnosis of primitive and preoperative lymph node staging [13]. Six studies investigated the diagnostic role of 18F-Fluciclovine PET/CT in detecting primary prostate cancer [14,15,16,17,18,19]. Three studies evaluated the role of lymph node staging [17,20,21] and six investigated the role of 18F-Fluociclovine in recurrence of prostate disease [13,22,23,24,25,26]. A study compared 18F-Fluciclovine PET-CT with bone scan in the evaluation of bone metastases [27]. 

### 3.2. Diagnostic Performance of 18F-Fluciclovine PET/CT in Different Clinical Setting

The diagnostic performance results of 18F-Fluciclovine PET/CT for the diagnosis of primary cancer, pre-operative LN staging, for prostate disease recovery, and for bone metastasis assessment are shown in Table 2. Pooled sensitivity and specificity were calculated according to a patient-based analysis. 

#### 3.2.1. Detection of Primary Prostate Cancer Lesion 

The per-patient pooled sensitivity for 18F-Fluciclovine PET/CT for the diagnosis of primary PCa was 0.83 (95% CI: 0.80–0.86), with I-square: 89.6% (Figure 2a) and a pooled of specificity of 0.77 (95% CI: 0.74–0.80) with I –square: 94.8% (Figure 2b). The likelihood ratio (LR) syntheses gave an overall LR+ of 5.22 (95% CI: 2.15–12.67) (Figure 2c) and LR– of 0.17 (95% CI: 0.8–0.38) (Figure 2d). The pooled DOR was 35.43 (95%CI: 6.66–188.55) (Figure 2e). The SROC curve indicates that the area under the cure was: 0.9224 (Figure 2f).

#### 3.2.2. Preoperative LN Staging

The per-patient pooled sensitivity for 18F-Fluciclovine PET/CT for preoperative LN staging was 0.57 (95% CI: 0.39–0.73), with I-square: 0.0% (Figure 3a) and a pooled specificity of 0.99 (95% CI: 0.94–1.00) with I –square: 0.0% (Figure 3b). Likelihood ratio (LR) syntheses gave an overall LR+ of 22.91 (95% CI: 5.67–92.65) (Figure 3c) and LR– of 0.48 (95% CI: 0.34–0.68) (Figure 3d). The pooled DOR was 52.55 (95% CI: 11.19–246.86) (Figure 3e). The SROC curve indicates that the area under the cure was 0.9410 (Figure 3f).

#### 3.2.3. Detection of Recurrent Disease

The per-patient pooled sensitivity for 18F-Fluciclovine PET/CT for the detection of recurrent disease was 0.68 (95% CI: 0.63–0.73), with I-square 91.8% (Figure 4a) and a pooled of specificity of 0.68 (95% CI: 0.60–0.75) with I-square 0.0% (Figure 4b). The likelihood ratio (LR) syntheses gave an overall LR+ of 2.43 (95% CI: 1.93–3.06) (Figure 4c) and LR– of 0.33 (95% CI: 0.20–0.54) (Figure 4d). The pooled DOR was 8.40 (95% CI: 5.13–13.75) (Figure 4e). The SROC curve indicates that the area under the cure was: 0.8086 (Figure 4f).

#### 3.2.4. Evaluation of Bone Metastases

It was not possible to perform a meta-analysis on bone metastases as there is only one study. 

#### 3.2.5. Evaluation of Heterogeneity in Meta-Regression Analysis

A heterogeneity in sensitivity was found among the studies presented (I-square > 20%). Therefore, the random-effect model was used as the most accurate model for all analyzes. The heterogeneity found is most likely due to the disparity in the number of populations included in the two or three study groups, particularly in preoperative LN staging. This is even more enveloped by the low number of studies reviewed, which was only three, and the detection of recurrent disease with a large number of false negatives reported in one study [20].

## 4. Discussion

18F-Fluciclovine is a synthetic amino acid used as a PET tracer for PCa. Cell uptake is a system mediated transport of neutral amino acids such as LAT 1 and ASCT2 across mammalian cell membranes, which are overexpressed in many cancer cells, including those from prostate cancer. Once inside the cell, 18F-Fluciclovine is not metabolized and is incorporated into proteins and only a small part is excreted in the urinary system. Similar to other tracers, the PET tracer 18F-Fluciclovine is a non-specific uptake by benign inflammatory prostatic tissue and in a variety of other malignancies. The aim of the study was to evaluate the diagnostic and specific sensitivity of 18F-Fluciclovine for the diagnosis of primary cancer, pre-operative LN staging, for the recovery of prostate disease and for the evaluation of bone metastases. In several studies, 18F-Fluciclovine PET/CT shows a higher uptake in intra-prostatic tumor foci than in normal prostate tissue (Figure 5a,b). However, 18F-Fluciclovine uptake in tumors is similar to that in BPH (benign prostate hyperplastic) nodules [23,26]. The Food and Drug Administration (FDA) and European Medicines Agency (EMA) have approved 18F-Fluciclovine in the recurrent setting only [28]. After definitive treatment for prostate cancer, patients are routinely followed up with serum PSA level. 18F-Fluciclovine is highly useful in the detection of recurrent prostate cancer, even in the presence of non-conclusive conventional imaging. In fact, CT or MRI scans may not detect or accurately characterize the biochemical relapse at the earliest stage [29]. However, functional imaging with Choline or Fluciclovine PET/CT associated with multi-parametric MRI (MP-MR) seems to be the most valuable technique in the detection of prostate cancer relapse [23]. These functional images are cost-effectiveness when PSA doubling time is short. 18F-Fluciclovine PET/CT shows detection rates of 72.0%, 83.3%, and 100% at PSA levels <1, 1–2. and >2 ng/mL, respectively [19]. In comparison with MP-MR, many studies concluded that 18F-Fluciclovine imaging for the evaluation of primary PCa was limited. Delayed imaging (20–28 min) could improve diagnostic performance for the characterization of primary cancer and can help guide biopsy in high-risk disease [17,23]. A study compared prospectively 18F-Fluciclovine and PSMA PET/CT scans for localizing recurrence of PCa after prostatectomy in patients with a PSA level <2.0 ng/mL. PSMA PET/CT detection rates for pelvic and extra-pelvic metastasis were higher than those for 18F-Flucicloviune PET/CT [10]. Other results, obtained in a more heterogeneous and at higher risk population, showed a better detection rate for 18F-Fluciclovine compared to PSMA for the detection of prostate bed recurrences in areas close to the bladder (37.9% and 27.6%, respectively) [30]. The lower urinary excretion of 18F-Flucicloviune PET compared to PSMA PET might be the explanation of this finding. However, PET/CT findings validation is not always feasible, especially in the recurrent setting. Nevertheless, it is well established that 18F-Fluciclovine is highly useful in the detection of recurrent prostate cancer, when conventional bone scan and CT and/or MRI imaging are negative [29]. Our meta-analysis study showed promising results in terms of sensitivity and specificity of 18F-Fluciclovine PET/CT, as recently reported in other meta-analysis recently published [31,32,33]. High specificity values have been observed for preoperative LN staging (almost 100%); acceptable (although lower) pooled specificity (68%) was obtained for the detection of PCa recurrence in terms of local recurrence and nodal localization. Discrepancy may be a consequence of a smaller number of studies included in meta-analysis of preoperative LN staging (which may have somehow reduced the statistical power of this sub-analysis) compared to the recurrent setting. The validation of positive findings still represents a challenge for medical imaging in oncology. In pre-surgery setting, a more accurate approach can be designed, and PET results can be validated by histology. Generally, lesion- or region-based validation is preferable and (especially for lymph node metastasis) positive PET lesions are compared with surgery templates. On the contrary, in the recurrent setting, the standard of truth is generally composite. Histological confirmation of metastatic sites is not often feasible due to ethical and practical reasons. Thus, PET findings are generally validated with informative conventional imaging that might have lower diagnostic accuracy compared to new generation imaging. Further validation can be obtained by complete PSA response in subjects treated with image-guided therapy. This heterogeneity might explain the different specificity observed in primary staging vs. recurrent setting. 

For the detection of bone metastases, further studies will be necessary, as the few studies considered are not able to provide statistically acceptable results. However, in this context, the physiological uptake of 18F-Fluciclovine might represent a limit. 

An accurate knowledge of normal and physiologic distribution and variants is important for proper interpretation of 18F-Fluciclovine PET/CT imaging. Uptake may also be present in benign conditions such as inflammation and infection, in other metabolically active benign lesions or in other malignancies. Nowadays, 18F-Fluciclovine is also being investigated for other cancer indications, such as brain metastases, gliomas and breast cancers. Similar to many other statistical techniques, meta-analysis is a powerful tool, but it needs to fulfill several key requirements to ensure the validity of its results. In our study, we defined objectives, including definitions of clinical variables, evaluation of risk of bias in the selection of studies and considering heterogeneity of the population.

## 5. Conclusions 

This systematic review and the related meta-analysis demonstrated promising results in term of sensitivity and specificity for 18F-Fluciclovine PET/CT to stage the primary lesion and in the assessment of nodal metastases, and for the detection of PCa locations in the recurrent setting. 18F-Fluciclovne PET can still be considered a valid new generation imaging procedure for staging PCa patients. However, the limited number of studies and the heterogeneity in the selected cohorts and different investigation protocols are limitations affecting the strength of these results.

## Figures and Tables

**Figure 1 diagnostics-11-00304-f001:**
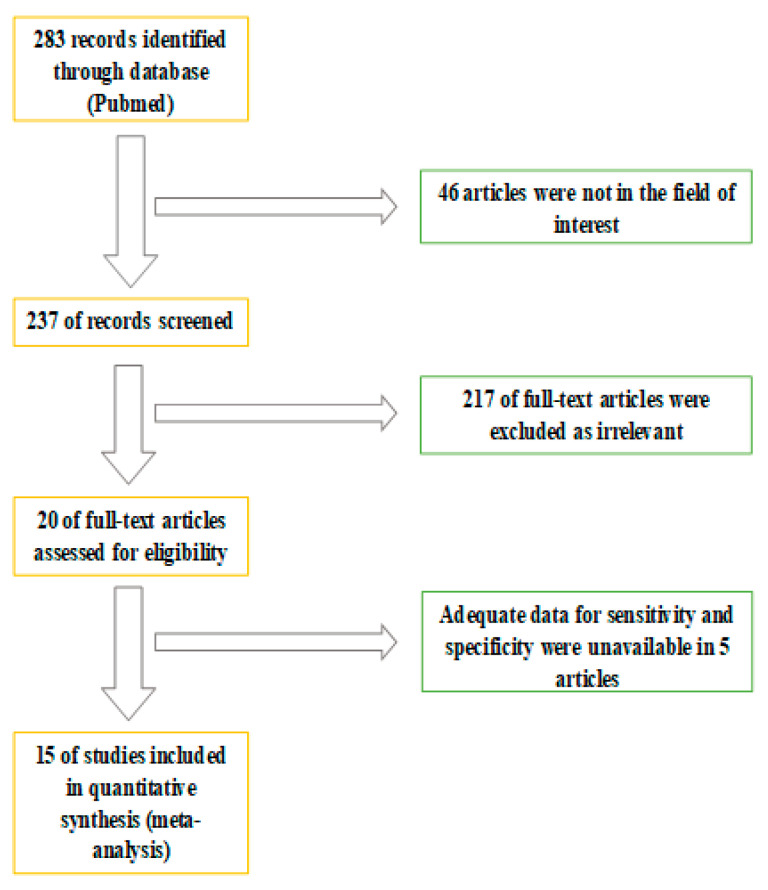
Flowchart of information through the different phases of the systematic review.

**Figure 2 diagnostics-11-00304-f002:**
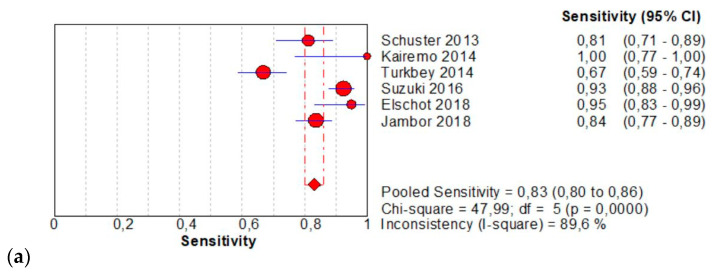
(**a**,**b**) Forest plot of 18F-Fluciclovine PET/CT for the diagnosis of prostate cancer (sensibility and specificity). (**c**,**d**) Forest plot of 18F-Fluciclovine PET/CT for the diagnosis of prostate cancer (positive and negative likelihood ratio). (**e**,**f**): Pooled DOR and SROC curve of 18F-luciclovine PET/CT for the diagnosis of prostate cancer.

**Figure 3 diagnostics-11-00304-f003:**
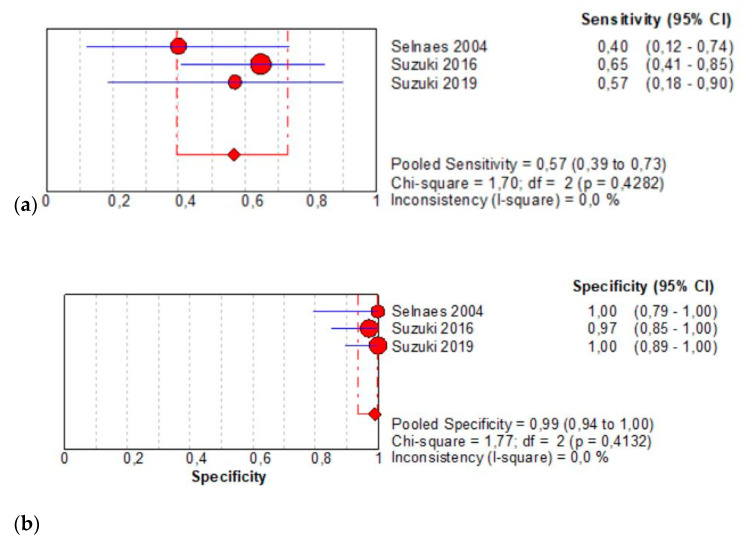
(**a**,**b**) Forest plot of 18F-Fluciclovine PET/CT for preoperative LN staging (sensibility and specificity). (**c**,**d**) Forest plot of 18F-Fluciclovine PET/CT for preoperative LN staging (positive and negative likelihood ratio). (**e**,**f**) Pooled DOR and SROC curve of 18F-luciclovine PET/CT for preoperative LN staging.

**Figure 4 diagnostics-11-00304-f004:**
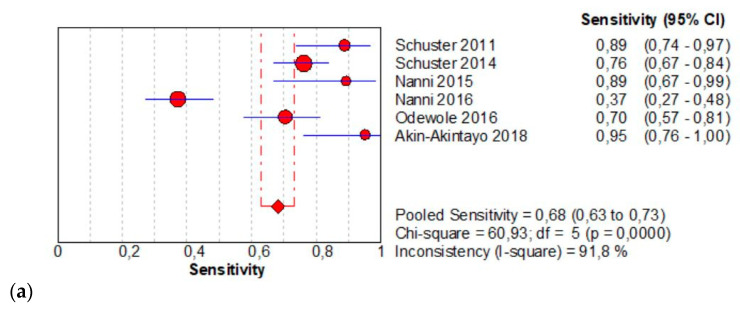
(**a**,**b**) Forest plot of 18F-Fluciclovine PET/CT for the detection of recurrent disease (sensibility and specificity). (**c**,**d**) Forest plot of 18F-Fluciclovine PET/CT for the detection of recurrent disease (positive and negative likelihood ratio). (**e**,**f**) Pooled DOR and SROC curve of 18F-luciclovine PET/CT for the detection of recurrent disease.

**Figure 5 diagnostics-11-00304-f005:**
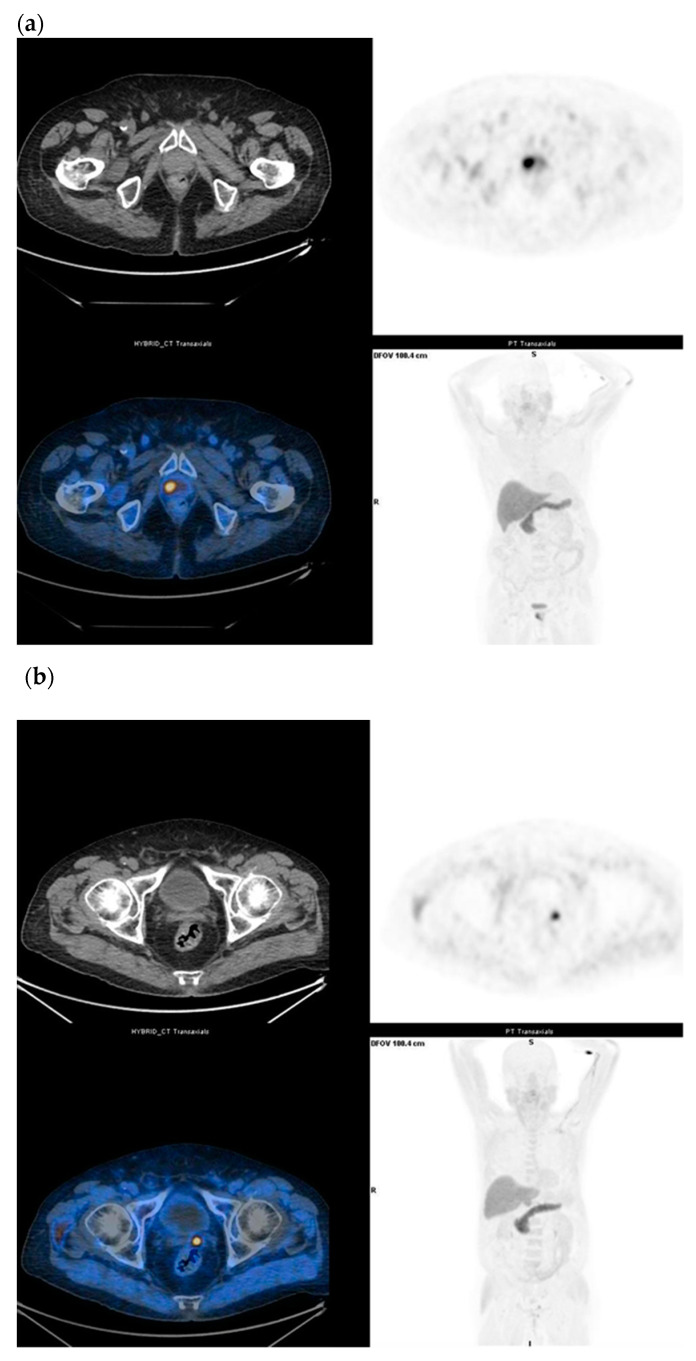
(**a**) Staging in un-operated prostate cancer under drug treatment (Gleason Score: 4 + 3) and with a moderate increase in serum PSA (1.67 ng/mL); 18F-Fluciclovine PET/CT shows intense parenchymal uptake (SUV_max_ 15) in the right paramedian site of the prostate. (**b**) Biochemical recurrence in prostate cancer (Gleason Score: 4 + 4) subjected to radical HIFU treatment three years ago and with PSA elevation (3.3 ng/mL); 18F-Fluciclovine PET/CT shows a focal uptake in the left prostatic lodge (SUV _max_ 7).

**Table 1 diagnostics-11-00304-t001:** Flow-chart for risk of bias and applicability concerns in Quality Assessment of Diagnostic Accuracy Studies (QUADAS-2).

Domain	Patient Selection	Index Test	Reference Standard	Flow and Timing
**Description**	Describe methods of patient selection. Describe included patients (previous testing, presentation, intended use of index test and setting)	Describe the index test and how it was conducted and interpreted	Describe the reference standard and how it was conducted and interpreted	Describe any patients who did not receive the index tests or reference standard or who were excluded from the 2 × 2 table. Describe the interval and any interventions between index tests and the reference standard
**Signaling questions** **(yes, no, unclear)**	Was a consecutive or random sample of patients enrolled? Was a case-control design avoided? Did the study avoid inappropriate exclusions?	Were the index test results interpreted without knowledge of the results of the reference standard? If a threshold was used, was it pre-specified?	Is the reference standard likely to correct classify the target condition? Were the reference standard results interpreted without knowledge of the results of the index test?	Was there an appropriate interval between index tests and reference standard? Did all patients receive a reference standard? Did all patients receive the same reference standard? Were all patients included in the analysis?
**Risk of bias**	Could the selection of patients have introduced bias?	Could the conduct or interpretation of the index test have introduced bias?	Could the reference standard, its conduct or its interpretation have introduced bias?	Could the patient flow have introduced bias?
**Concerns about applicability** **(high, low, unclear)**	Are there concerns that the included patients do not match the review question?	Are there concerns that the index tests, its conduct, or its interpretation differ from the review question?	Are there concerns that the target condition as defined by the reference standard does not match the review question?	

**Table 2 diagnostics-11-00304-t002:** Authors, patients, and study characteristics.

First Author, Year	True +	False +	False −	True −	Sensitivity (95% CI)	Specificity (95% CI)	No Pts	Age	PSA ng/mL	GS
**Diagnosis of primary PCa**										
Schuster 2013	65	20	15	20	0.81 (0.7–1.00)	0.50 (0.34–0.65)	10	60.8	8.2	6–10
Kairemo 2014	14	3	0	15	1.00 (0.77–1.00)	0.83 (0.59–0.96)	26	68.1	7.9	7.1
Turkbey 2014	99	99	49	173	0.67 (0.59–0.74)	0.64 (0.58–0.69)	22	62.2	13.5	6–9
Suzuki 2016	173	7	14	64	0.93 (0.88–0.96)	0.90 (0.81–0.96)	68	67.3	88.6	6–10
Elschot 2018	38	9	2	72	0.95 (0.83–0.99)	0.89 (0.79–0.95)	28	66	n.a.	n.a.
Jambor 2018	139	6	27	140	0.84 (0.77–0.89)	0.96 (0.91–0.98)	32	65	12.0	7
**Preoperative LN staging**										
Selnaes 2018	4	0	6	16	0.40 (0.12–0.73)	1.00 (0.79–1.00)	28	66.2	14.6	7–11
Suzuki 2016	13	1	7	34	0.65 (0.40–0.84)	0.97 (0.85–0.99)	68	67.3	88.6	6–10
Suzuki 2019	4	0	3	33	0.57 (0.18–0.90)	0.85 (0.68–0.95)	28	69	12.8	7–10
**Detection of recurrent** **disease**										
Schuster 2011	32	4	4	8	0.89 (0.74–0.97)	0.67 (0.35–0.90)	93	68.3	6.6	n.a.
Schuster 2014	77	19	24	41	0.76 (0.67–0.84)	0.68 (0.55–0.80)	28	68	9.8	7
Nanni 2015	17	10	2	21	0.89 (0.67–0.99)	0.68 (0.49–0.83)	50	67	3.2	n.a.
Nanni 2016	32	1	54	2	0.37 (0.27–0.48)	0.67 (0.09–0.99)	89	69	6.9	n.a.
Odewole 2016	43	7	18	24	0.70 (0.57–0.81)	0.77 (0.59–0.90)	53	67.5	7.2	n.a.
Akin-Akintayo 2018	20	9	1	10	0.95 (0.76–1.00)	0.53 (0.29–0.76)	24	70.8	8.5	7
**Detection of bone** **metastases**										
Chen Bo 2019	8	13	3	0	1 (0.77–1)	0.98 (0.92–1)	26	70	1.3	6–10

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
