# Peer review of "18F-Fluciclovine Positron Emission Tomography in Prostate Cancer: A Systematic Review and Diagnostic Meta-Analysis"

_diagnostics, 2021, doi:10.3390/diagnostics11020304_

Round 1

Reviewer 1 Report

Dear authors,

I think that this meta-analysis is interesting and the topic is surely of clinical interest.

The potential role of 18F-FACBC PET/CT in prostate cancer, is still under investigation and only initial promising results and recent evidences are present in literature. This restricted field of research is clearly represented also by the output of the systematic review.

However, in the paper there are some points to improve and correct to let the paper more clear and complete.

After these improvement and corrections, I think that the article could be accepted.

COMMENTS

  1. Abstract

- L22: LN check the acronym.

  1. References

- Please check the cross references of the whole article: text/bibliography, table/bibliography, graph/bibliography.

In the next points the explanations.

  1. Material and methods

- L79-85: Since the title is: “Role of 18F-Fluciclovine PET/CT in Prostate Cancer: A Systematic Review and Diagnostic Meta-Analysis”, include PET alone studies represents a potential bias.

- L84: please add a table for QUADAS2

  1. TABLE 1

- Please, don’t use in the whole table the capital letters and check the use of the bold, add the number of reference near to the name/year of the first author.

- Re-check the whole table: names, references and content of the studies.

Example:

L98-99 Three studies evaluated the role of lymph node staging [13,19,20]

  1. Odewole, O.A.; Tade, F.I.; Nieh, P.T.; Savir-Baruch, B.; Jani, A.B.; Master, V.A.; Rossi, P.J.; Halkar, R.K.; Osunkoya, A.O.; Akin-287 Akintayo, O.; Zhang, C.; Chen, Z.; Goodman, M.M.; Schuster; D.M. Recurrent prostate cancer detection with anti-3-288 [18F]FACBC PET/CT: comparison with CT. Eur J Nucl Med Mol Imaging 2016, 43:1773-1783.
  2. Nanni, C.; Zanoni, L.; Pultrone, C.; Schiavina, R.; Brunocilla, E.; Lodi, F.; Malizia, C.; Ferrari, M.; Rigatti, P.; Fonti, C.; Martorana, 304 G.; Fanti, S. 18F-FACBC (anti1-amino-3-18F-fluorocyclobutane-1-carboxylic acid) versus 11C-choline PET/CT in prostate cancer 305 relapse: results of a prospective trial. Eur J Nucl Med Mol Imaging 2016, 43:1601-1610.. 306
  3. Schuster, D.M.; Savir-Baruch, B.; Nieh, P.T.; Master, V.A.; Halkar, R.K.; Rossi, P.J.; Lewis, M.M.; Nye, J.A.; Yu, W.; Bowman, 307 F.D.; Goodman, M.M. Detection of recurrent prostate carcinoma with anti-1-amino-3-18F-fluorocyclobutane-1-carboxylic acid 308 PET/CT and 111In-capromab pendetide SPECT/CT. Radiology 2011, 259, 852-861.

Are they?

  1. Selnæs, K.M.; Kruger-Stokke B.; Elschot, M.; Willoch, F.; Størkersen, Ø.; Sandsmark, E.; Moestue, S.A.; Tessem , M.; Hal-vorsen, D.; Kjøbli, E.; Angelsen, A.; Langørgen, S.; Bertilsson, H.; Bathen, T.F. 18F-Fluciclovine PET/MRI for preoperative lymph node staging in high-risk prostate cancer patients. Eur Radiol 2018, 28, 3151-3159. 324
  2. Suzuki, H.; Inoue, Y.; Fujimoto, H.; Yonese, J.; Tanabe, K.; Fukasawa, S.; Inoue, T.; Saito, S.; Ueno, M.; Otaka, A. Diagnostic performance and safety of NMK36 (trans-1-amino-3-[18F]fluorocyclobutanecarboxylic acid)- PET/CT in primary prostate cancer: multicenter Phase IIb clinical trial. Jpn J Clin Oncol 2016, 46, 152-162.
  3. Suzuki, H.; Jinnouchi, S.; Kaji, Y.; Kishida, T.; Kinoshita, H.; Yamaguchi, S.; Tobe, T.; Okamura, T.; Kawakita, M.; Furukawa, J.; Otaka, A.; Kakehi, Y. Diagnostic performance of 18F-fluciclovine PET/CT for regional lymph node metastases in patients with primary prostate cancer: a multicenter phase II clinical trial. Jpn J Clin Oncol 2019, 49, 803-811.

PREOPERATIVE LN STAGING

SELNAES 2004

4

0

6

16

0.40 (0.12-0.73)

1.00 (0.79-1.00)

28

66.2

14.6

7-11

SUZUKI 2016

13

1

7

34

0.65 (0.40-0.84)

0.97 (0.85-0.99)

68

67.3

88.6

6-10

SUZUKI 2019

4

0

3

33

0.57 (0.18-0.90)

0.85 (0.68-0.95)

28

69

12.8

7-10

- Check the year or the reference of SELNAES 2004 (?) 

  1. Discussion

- L.202-203 : In fact, CT or MRI scans may not detect or accurately characterize the biochemical relapse at earliest.

Add references to support the sentence.

- L222-223 : Our meta-analysis study showed promising results in terms of sensitivity and specificity of 18F-Fluciclovine PET/CT and comparable to other similar studies  [31].

Searching in literature, recent meta-analyses are not cited. Expand the comparison and describe the results, please add:

  1. Laudicella, R; Albano, D; Alongi, P; et al. (18) F-Facbc in Prostate Cancer: A Systematic Review and Meta-Analysis. Cancers 2019, 11, 1348.
  2. Kim, S-J; Lee, S. The role of (18) F-fluciclovine PET in the management of prostate cancer: A systematic review and meta-analysis. Clin. Radiol. 2019, 74, 886–892.

Author Response

Answer to Reviewers’ Comments

Dear Editor,

We would thank you for giving us the chance to review and improve our manuscript. All reviewers’ comments have been taken in consideration. A point-by-point revision was performed and all changes have been highlighted in red font in the text.

All authors approved all changes performed and reviewed the final version of the mansuscript.

Dr. Cinzia Romagnolo, MD

Reviewer #1

R1: Dear authors, I think that this meta-analysis is interesting and the topic is surely of clinical interest. The potential role of 18F-FACBC PET/CT in prostate cancer, is still under investigation and only initial promising results and recent evidences are present in literature. This restricted field of research is clearly represented also by the output of the systematic review. However, in the paper there are some points to improve and correct to let the paper more clear and complete. After these improvement and corrections, I think that the article could be accepted.

R1-1: Abstract. L22: LN check the acronym.

A: done

R1-2: References. Please check the cross references of the whole article: text/bibliography, table/bibliography, graph/bibliography.

A: we would thank the reviewer for his/her comment. The reference list has been double checked and improved according to the reviewer’s comment.

R1-3: Material and methods. - L79-85: Since the title is: “Role of 18F-Fluciclovine PET/CT in Prostate Cancer: A Systematic Review and Diagnostic Meta-Analysis”, include PET alone studies represents a potential bias.

A: we would thank the reviewer for the comment. To the best of our knowledge, only study involving PET hybrid tomographs have been considered. However, to eventually avoid misinterpretation the title has been rephrase accordingly, as follow: “18F-Fluciclovine Positron Emission Tomography in Prostate Cancer: A Systematic Review and Diagnostic Meta-Analysis”

R1-4: L84: please add a table for QUADAS2

A: Done. The table for QUADAS is now reported as Table 1.

R1-5: TABLE 1. Please, don’t use in the whole table the capital letters and check the use of the bold, add the number of reference near to the name/year of the first author.

A: done.

R1-6: Re-check the whole table: names, references and content of the studies.

A: done.  

R1-7: L98-99 Three studies evaluated the role of lymph node staging [13,19,20]

Odewole, O.A.; Tade, F.I.; Nieh, P.T.; Savir-Baruch, B.; Jani, A.B.; Master, V.A.; Rossi, P.J.; Halkar, R.K.; Osunkoya, A.O.; Akin-287 Akintayo, O.; Zhang, C.; Chen, Z.; Goodman, M.M.; Schuster; D.M. Recurrent prostate cancer detection with anti-3-288 [18F]FACBC PET/CT: comparison with CT. Eur J Nucl Med Mol Imaging 2016, 43:1773-1783.

Nanni, C.; Zanoni, L.; Pultrone, C.; Schiavina, R.; Brunocilla, E.; Lodi, F.; Malizia, C.; Ferrari, M.; Rigatti, P.; Fonti, C.; Martorana, 304 G.; Fanti, S. 18F-FACBC (anti1-amino-3-18F-fluorocyclobutane-1-carboxylic acid) versus 11C-choline PET/CT in prostate cancer relapse: results of a prospective trial. Eur J Nucl Med Mol Imaging 2016, 43:1601-1610.

Schuster, D.M.; Savir-Baruch, B.; Nieh, P.T.; Master, V.A.; Halkar, R.K.; Rossi, P.J.; Lewis, M.M.; Nye, J.A.; Yu, W.; Bowman, F.D.; Goodman, M.M. Detection of recurrent prostate carcinoma with anti-1-amino-3-18F-fluorocyclobutane-1-carboxylic acid 308 PET/CT and 111In-capromab pendetide SPECT/CT. Radiology 2011, 259, 852-861.

 Are they?

Selnæs, K.M.; Kruger-Stokke B.; Elschot, M.; Willoch, F.; Størkersen, Ø.; Sandsmark, E.; Moestue, S.A.; Tessem , M.; Hal-vorsen, D.; Kjøbli, E.; Angelsen, A.; Langørgen, S.; Bertilsson, H.; Bathen, T.F. 18F-Fluciclovine PET/MRI for preoperative lymph node staging in high-risk prostate cancer patients. Eur Radiol 2018, 28, 3151-3159. 324

Suzuki, H.; Inoue, Y.; Fujimoto, H.; Yonese, J.; Tanabe, K.; Fukasawa, S.; Inoue, T.; Saito, S.; Ueno, M.; Otaka, A. Diagnostic performance and safety of NMK36 (trans-1-amino-3-[18F]fluorocyclobutanecarboxylic acid)- PET/CT in primary prostate cancer: multicenter Phase IIb clinical trial. Jpn J Clin Oncol 2016, 46, 152-162.

Suzuki, H.; Jinnouchi, S.; Kaji, Y.; Kishida, T.; Kinoshita, H.; Yamaguchi, S.; Tobe, T.; Okamura, T.; Kawakita, M.; Furukawa, J.; Otaka, A.; Kakehi, Y. Diagnostic performance of 18F-fluciclovine PET/CT for regional lymph node metastases in patients with primary prostate cancer: a multicenter phase II clinical trial. Jpn J Clin Oncol 2019, 49, 803-811.

A: The numbering of the references was incorrect. An adequate check has been carried out and now the numbering has been corrected.

R1-8: PREOPERATIVE LN STAGING: SELNAES 2004, SUZUKI 2016, SUZUKI 2019. Check the year or the reference of SELNAES 2004 (?)

A: done: SELNAES, 2018

 R1-9: Discussion, L.202-203 : In fact, CT or MRI scans may not detect or accurately characterize the biochemical relapse at earliest. Add references to support the sentence.

A: The following reference has been added:  De Visschere PJL, Standaert C, Fütterer JJ, et al. A Systematic Review on the Role of Imaging in Early Recurrent Prostate Cancer. Eur Urol Oncol. 2019 Feb;2(1):47-76.

R1-10: L222-223 : Our meta-analysis study showed promising results in terms of sensitivity and specificity of 18F-Fluciclovine PET/CT and comparable to other similar studies  [31]. Searching in literature, recent meta-analyses are not cited. Expand the comparison and describe the results, please add:

  • Laudicella, R; Albano, D; Alongi, P; et al. (18) F-Facbc in Prostate Cancer: A Systematic Review and Meta-Analysis. Cancers 2019, 11, 1348.
  • Kim, S-J; Lee, S. The role of (18) F-fluciclovine PET in the management of prostate cancer: A systematic review and meta-analysis. Clin. Radiol. 2019, 74, 886–892.

A: We agree with reviewer’s comment and the above mentioned references have been added. The text has been amended accordingly, as follow: “Our meta-analysis study showed promising results in terms of sensitivity and specificity of 18F-Fluciclovine PET/CT, as recently reported in other meta-analysis recently published”.

Reviewer #2

R2: This systematic review and metanalysis aims to explore the diagnostic accuracy of Fluciclovine PET in prostate cancer, both considering primary staging prior to radical therapy, biochemical recurrence, and the advanced setting. The study covers an emerging and interesting issue. It is conducted with appropriate methodology and interpretation of the data. I just propose a few minor suggestions to improve its readability.

R2-1: Results and discussion should be clearly separated into two different chapters.

A: The text has been amended according to reviewer comment and the two sections have been separated.

R2-2: paragraph 3.2.2 “The per-patient pooled sensitivity for 18F-fluciclovine PET/CT for diagnosis of primary PCa was 0.68 (95% CI: 0.63-0.73), with I-square: 91,8% (Fig. 4a) and a pooled of specificity of 0.68 (95% CI:0.60-0.75) with I –square: 0,0% (Fig. 4b).” I am not able to understand why the authors reported the accuracy for diagnosis of primary PCa instead of recurrency (which is the topic of the paragraph).

A: we would thank the reviewer for this comment. It was a typo and clearly the paragraph refers to recurrent setting only. The text has been changed accordingly, as follow: ”The per-patient pooled sensitivity for 18F-Fluciclovine PET/CT for detection of recurrent disease was 0.68”.

R2-3: Looking at the meta-analysis results, a relevant mismatch can be observed regarding specificity from the pre-treatment setting to the setting of recurrence. FACBC specificity is almost 100% in the former situation, while in the latter, pooled specificity is below 70%. This intriguing result merits to be commented on in the discussion. Is it related to the occurrence of inflammation-related false-positive findings typical of the post-treatment setting?

A: we thank the reviewer for his/her comment. The validation of positive findings always represents a challenge for medical imaging in oncology. In pre-surgery setting, a more accurate approach can be designed, and PET results can be validated by histology. Generally, lesion- or region-based validation is preferable and (especially for lymph node metastasis) positive PET lesions are compared with surgery templates. On the contrary, in the recurrent setting the standard of truth is generally composite. Histological confirmation of metastatic sites is not often feasible due to ethical and practical reasons. Thus, PET findings are generally validated with informative conventional imaging that might have lower diagnostic accuracy compared to new generation imaging. Further validation can be obtained by complete PSA response in subjects treated with image-guided therapy. This heterogeneity might explain the different specificity observed in primary staging vs. recurrent setting.  

We implemented this comment in the discussion section, as reported below: “High specificity values have been observed for preoperative LN staging (almost 100%); acceptable (although lower) pooled specificity (68%) was obtained for the detection of PCa recurrence in terms of local recurrence and nodal localization. Discrepancy may be a consequence of a smaller number of studies included in me-ta-analysis of preoperative LN staging (which may have reduced somehow the statis-tical power of this sub-analysis) compared to the recurrent setting. The validation of positive findings still represents a challenge for medical imaging in oncology. In pre-surgery setting, a more accurate approach can be designed, and PET results can be validated by histology. Generally, lesion- or region-based validation is preferable and (especially for lymph node metastasis) positive PET lesions are compared with surgery templates. On the contrary, in the recurrent setting the standard of truth is generally composite. Histological confirmation of metastatic sites is not often feasible due to ethical and practical reasons. Thus, PET findings are generally validated with informa-tive conventional imaging that might have lower diagnostic accuracy compared to new generation imaging. Further validation can be obtained by complete PSA response in subjects treated with image-guided therapy. This heterogeneity might explain the different specificity observed in primary staging vs. recurrent setting”.

R2-4: This is not the first systematic review and meta-analysis on this topic. The authors already cited the one by Bin et al. (PMID: 31998634). However, also other meta-analyses have been previously published (PMID: 31514479, PMID: 25907118). The authors should discuss the differences between their study and the others which are already available.

A: We thank the reviewer for this insight. As stated previously (Rev 1, comment 10) two additional references has been added. In this meta-analysis we reported about sensitivity and specificity of Fluciclovine PET in different clinical settings, thus representing the main novelty of this systematic review. We implemented this comment in the discussion section: “Our meta-analysis study showed promising results in terms of sensitivity and specificity of 18F-Fluciclovine PET/CT, as recently reported in other meta-analysis recently published.”

Reviewer 2 Report

This systematic review and metanalysis aims to explore the diagnostic accuracy of Fluciclovine PET in prostate cancer, both considering primary staging prior to radical therapy, biochemical recurrence, and the advanced setting.

The study covers an emerging and interesting issue. It is conducted with appropriate methodology and interpretation of the data. I just propose a few minor suggestions to improve its readability.

1) Results and discussion should be clearly separated into two different chapters.

2) paragraph 3.2.2 “The per-patient pooled sensitivity for 18F-fluciclovine PET/CT for diagnosis of primary PCa was 0.68 (95% CI: 0.63-0.73), with I-square: 91,8% (Fig. 4a) and a pooled of specificity of 0.68 (95% CI:0.60-0.75) with I –square: 0,0% (Fig. 4b).” I am not able to understand why the authors reported the accuracy for diagnosis of primary PCa instead of recurrency (which is the topic of the paragraph).

3) Looking at the meta-analysis results, a relevant mismatch can be observed regarding specificity from the pre-treatment setting to the setting of recurrence. FACBC specificity is almost 100% in the former situation, while in the latter, pooled specificity is below 70%. This intriguing result merits to be commented on in the discussion. Is it related to the occurrence of inflammation-related false-positive findings typical of the post-treatment setting?

4) This is not the first systematic review and meta-analysis on this topic. The authors already cited the one by Bin et al. (PMID: 31998634). However, also other meta-analyses have been previously published (PMID: 31514479, PMID: 25907118). The authors should discuss the differences between their study and the others which are already available.

Author Response

Answer to Reviewers’ Comments

Dear Editor,

We would thank you for giving us the chance to review and improve our manuscript. All reviewers’ comments have been taken in consideration. A point-by-point revision was performed and all changes have been highlighted in red font in the text.

All authors approved all changes performed and reviewed the final version of the mansuscript.

Dr. Cinzia Romagnolo, MD

Reviewer #2

R2: This systematic review and metanalysis aims to explore the diagnostic accuracy of Fluciclovine PET in prostate cancer, both considering primary staging prior to radical therapy, biochemical recurrence, and the advanced setting. The study covers an emerging and interesting issue. It is conducted with appropriate methodology and interpretation of the data. I just propose a few minor suggestions to improve its readability.

R2-1: Results and discussion should be clearly separated into two different chapters.

A: The text has been amended according to reviewer comment and the two sections have been separated.

R2-2: paragraph 3.2.2 “The per-patient pooled sensitivity for 18F-fluciclovine PET/CT for diagnosis of primary PCa was 0.68 (95% CI: 0.63-0.73), with I-square: 91,8% (Fig. 4a) and a pooled of specificity of 0.68 (95% CI:0.60-0.75) with I –square: 0,0% (Fig. 4b).” I am not able to understand why the authors reported the accuracy for diagnosis of primary PCa instead of recurrency (which is the topic of the paragraph).

A: we would thank the reviewer for this comment. It was a typo and clearly the paragraph refers to recurrent setting only. The text has been changed accordingly, as follow: ”The per-patient pooled sensitivity for 18F-Fluciclovine PET/CT for detection of recurrent disease was 0.68”.

R2-3: Looking at the meta-analysis results, a relevant mismatch can be observed regarding specificity from the pre-treatment setting to the setting of recurrence. FACBC specificity is almost 100% in the former situation, while in the latter, pooled specificity is below 70%. This intriguing result merits to be commented on in the discussion. Is it related to the occurrence of inflammation-related false-positive findings typical of the post-treatment setting?

A: we thank the reviewer for his/her comment. The validation of positive findings always represents a challenge for medical imaging in oncology. In pre-surgery setting, a more accurate approach can be designed, and PET results can be validated by histology. Generally, lesion- or region-based validation is preferable and (especially for lymph node metastasis) positive PET lesions are compared with surgery templates. On the contrary, in the recurrent setting the standard of truth is generally composite. Histological confirmation of metastatic sites is not often feasible due to ethical and practical reasons. Thus, PET findings are generally validated with informative conventional imaging that might have lower diagnostic accuracy compared to new generation imaging. Further validation can be obtained by complete PSA response in subjects treated with image-guided therapy. This heterogeneity might explain the different specificity observed in primary staging vs. recurrent setting.  

We implemented this comment in the discussion section, as reported below: “High specificity values have been observed for preoperative LN staging (almost 100%); acceptable (although lower) pooled specificity (68%) was obtained for the detection of PCa recurrence in terms of local recurrence and nodal localization. Discrepancy may be a consequence of a smaller number of studies included in me-ta-analysis of preoperative LN staging (which may have reduced somehow the statis-tical power of this sub-analysis) compared to the recurrent setting. The validation of positive findings still represents a challenge for medical imaging in oncology. In pre-surgery setting, a more accurate approach can be designed, and PET results can be validated by histology. Generally, lesion- or region-based validation is preferable and (especially for lymph node metastasis) positive PET lesions are compared with surgery templates. On the contrary, in the recurrent setting the standard of truth is generally composite. Histological confirmation of metastatic sites is not often feasible due to ethical and practical reasons. Thus, PET findings are generally validated with informa-tive conventional imaging that might have lower diagnostic accuracy compared to new generation imaging. Further validation can be obtained by complete PSA response in subjects treated with image-guided therapy. This heterogeneity might explain the different specificity observed in primary staging vs. recurrent setting”.

R2-4: This is not the first systematic review and meta-analysis on this topic. The authors already cited the one by Bin et al. (PMID: 31998634). However, also other meta-analyses have been previously published (PMID: 31514479, PMID: 25907118). The authors should discuss the differences between their study and the others which are already available.

A: We thank the reviewer for this insight. As stated previously (Rev 1, comment 10) two additional references has been added. In this meta-analysis we reported about sensitivity and specificity of Fluciclovine PET in different clinical settings, thus representing the main novelty of this systematic review. We implemented this comment in the discussion section: “Our meta-analysis study showed promising results in terms of sensitivity and specificity of 18F-Fluciclovine PET/CT, as recently reported in other meta-analysis recently published.”